# Phosphorylation-Dependent Differences in CXCR4-LASP1-AKT1 Interaction between Breast Cancer and Chronic Myeloid Leukemia

**DOI:** 10.3390/cells9020444

**Published:** 2020-02-14

**Authors:** Elke Butt, Katrin Stempfle, Lorenz Lister, Felix Wolf, Marcella Kraft, Andreas B. Herrmann, Cristina Perpina Viciano, Christian Weber, Andreas Hochhaus, Thomas Ernst, Carsten Hoffmann, Alma Zernecke, Jochen J. Frietsch

**Affiliations:** 1Institute of Experimental Biomedicine, University Hospital Wuerzburg, Josef-Schneider-Straße 2, 97080 Wuerzburg, Germany; katrin.stempfle@gmx.de (K.S.); orenz@lister-esw.de (L.L.); felix.wolf96@icloud.com (F.W.); marcella.kraft@gmx.de (M.K.); andreasherrmann1995@gmail.com (A.B.H.); Zernecke_A@ukw.de (A.Z.); 2Institute of Molecular Cell Biology, CMB-Center for Molecular Biomedicine, University Hospital Jena, Hans-Knöll-Straße 2, 07745 Jena, Germany; cristina.perpina_viciano@uni-wuerzburg.de (C.P.V.); Carsten.Hoffmann@med.uni-jena.de (C.H.); 3Rudolf Virchow Center for Experimental Biomedicine, University of Wuerzburg, Josef-Schneider-Str. 5, 97080 Wuerzburg, Germany; 4Institute for Cardiovascular Prevention, LMU Munich, 80336 Munich, Germany; ipek.office@med.lmu.de; 5Cardiovascular Research Institute Maastricht, Department of Biochemistry, Maastricht University, 6229 ER Maastricht, The Netherlands; 6DZHK (German Centre for Cardiovascular Research), partner site Munich Heart Alliance, 80802 Munich, Germany; 7Klinik für Innere Medizin II, Abteilung für Hämatologie und internistische Onkologie, Universitätsklinikum Jena, Am Klinikum 1, 07747 Jena, Germany; Andreas.Hochhaus@med.uni-jena.de (A.H.); Thomas.Ernst@med.uni-jena.de (T.E.); Jochen.Frietsch@med.uni-jena.de (J.J.F.)

**Keywords:** LASP1, CXCR4, AKT1, CML, breast cancer

## Abstract

The serine/threonine protein kinase AKT1 is a downstream target of the chemokine receptor 4 (CXCR4), and both proteins play a central role in the modulation of diverse cellular processes, including proliferation and cell survival. While in chronic myeloid leukemia (CML) the CXCR4 is downregulated, thereby promoting the mobilization of progenitor cells into blood, the receptor is highly expressed in breast cancer cells, favoring the migratory capacity of these cells. Recently, the LIM and SH3 domain protein 1 (LASP1) has been described as a novel CXCR4 binding partner and as a promoter of the PI3K/AKT pathway. In this study, we uncovered a direct binding of LASP1, phosphorylated at S146, to both CXCR4 and AKT1, as shown by immunoprecipitation assays, pull-down experiments, and immunohistochemistry data. In contrast, phosphorylation of LASP1 at Y171 abrogated these interactions, suggesting that both LASP1 phospho-forms interact. Finally, findings demonstrating different phosphorylation patterns of LASP1 in breast cancer and chronic myeloid leukemia may have implications for CXCR4 function and tyrosine kinase inhibitor treatment.

## 1. Introduction

The LIM and SH3 protein 1 (LASP1) was first identified in human metastatic lymph nodes in breast cancer patients [1]. The *LASP1* gene is located on chromosome 17q12 and encodes for a protein with an N-terminal LIM domain, followed by two nebulin-like repeats, a linker region with two phosphorylation sites at S146 and Y171, and a C-terminal SH3 domain, known to bind to proline-rich proteins like lipoma-preferred partner (LPP), zyxin, dynamin, vimentin, and zona occludens protein 2 (ZO2) [2]. Phosphorylation of LASP1 at S146 by protein kinase A or protein kinase G mainly attenuates binding to filamentous F-actin, zyxin, and lipoma protein partner (LPP), thus allowing subcellular relocalization of LASP1, [3] while phosphorylation at Y171 by c-Src and c-ABL non-receptor tyrosine kinases is associated with cell spreading [4] and apoptosis [5]. For the cysteine-rich LIM domain of LASP1, binding to the chemokine receptors 1-4 (CXCR1-4) at their carboxy-termini has been described [6]. While the interaction with CXCR1-3 is independent of LASP1 phosphorylation, binding to CXCR4 requires LASP1 phosphorylation at S146 [6]. CXCR4 is mainly known for its crucial role in the homing and retention of hematopoietic stem and progenitor cells in stem cell niches of the bone marrow [7]. In chronic myeloid leukemia (CML), CXCR4 expression is down-regulated by the fusion protein BCR-ABL, and associated with a defective adhesion of CML cells to bone marrow stroma [8].

In this regard, LASP1 has been identified as member of a six genes signature highly predictive for CML [9] and a role of LASP1-CXCR4 in CML progression is discussed [10]. In addition, the constitutively active tyrosine kinase leads to hyperphosphorylation of proteins like Crk-like protein (CRKL) and LASP1 [11].

Although LASP1 was originally identified as a structural cytoskeletal and adaptor protein [12], an overexpression of LASP1 has been reported in numerous tumor entities [13] and recent data have also provided evidence for its transcriptional activity [14,15].

In breast cancer, CXCR4 promotes metastasis to organs like bone, liver, and lung, sites commonly affected by metastatic breast cancer, where its ligand, C-X-C motif chemokine 12 (CXCL12), is expressed in large quantities [16,17]. In addition, activation of CXCR4 in breast cancer cells facilitates nuclear translocation of LASP1 and an association of the protein with the transcription factor Snail, associated with epithelial-to-mesenchymal transition (EMT), and the protein complex UHRF1-DNMT1-G9a, involved in epigenetic modulation, is observed [14].

CXCR4 mediates intracellular signaling through a classical heterotrimeric G-protein, composed of Gα_i_, Gβ, and Gγ subunits. The Gα_i_ monomer inhibits adenylyl cyclase activity and triggers MAPK and PI3K pathway activation [18], whereas the Gβγ dimer induces intracellular calcium mobilization through the activation of phospholipase C. Recent evidence also points towards an influence of LASP1 on the PI3K/AKT pathway, one of the most frequently dysregulated signals in cancer [19]. This pathway is initiated by receptor tyrosine kinase (RTK) or G-protein coupled receptor activation, like CXCR4, thus inducing phosphorylation of PIP2 to PIP3 by PI3K, thereby recruiting AKT1 and phosphoinositide-dependent kinase (PDK1) to the plasma membrane, where AKT1 is phosphorylated by PDK1 at T308. Full activation requires additional AKT1 phosphorylation at S473 by the mTOR2 complex. The process is negatively regulated by phosphatases, either by direct dephosphorylation of AKT1, or by PTEN (phosphatase and tensin homolog) converting PIP3 back to PIP2 [20].

Decreased pAKT1-S473 phosphorylation after LASP1 depletion has been observed before in several tumor cell lines, glioblastoma [21], gall bladder [22] and nasopharyngeal carcinoma [23], while LASP1 overexpression induces phosphorylation in colorectal cancer cells [24] and non-small cell lung cancer [25]. However, the underlying mechanisms are still controversial. Shao et al. suggested a molecular mechanism by which LASP1 and COPS5 (COP9 signalosome subunit 5) synergistically stimulate ubiquitination and degradation of 14-3-3, indirectly causing AKT1-S473 phosphorylation [24]. A recent publication by the same group proposed that LASP1 promotes ubiquitination and degradation of PTEN, thus enhancing PIP2 phosphorylation to PIP3 and a concomitant AKT activation [23].

In this study, we demonstrate a direct interaction of LASP1 with AKT1. In addition, we investigate the differences in LASP1-CXCR4 signaling in breast cancer and CML, as these two tumor entities display different roles for CXCR4, despite comparable overexpression of the receptor binding-partner LASP1. The results prompted us to propose two connected models, explaining the variations in tumor progression in solid breast cancer and hematopoietic malignancy based on differences in LASP1 phosphorylation.

## 2. Materials and Methods

### 2.1. Plasmid Constructs and Mutagenesis

pcDNA3-HA-AKT1 full length (Addgene, Cambridge, MA, USA, plasmid #73408) and pCDNA3-HA-AKT1-aa1-408 (Addgene, plasmid #73412) were a gift from Chen [26]. Point mutations within AKT1 sequence (L420S, L426S, and R465S) were performed using the Agilent QuickChange XL Site-directed mutagenesis kit (Agilent, Santa Clara, CA, USA).

Generation of MDAMB-231 stably transfected with shRNA LASP1 was described earlier [15].

Full-length cDNA for human LASP1 (GenBank accession number X82456) was cloned into the pGEX4T1 vector (GE Healthcare, Freiburg, Germany). For the generation of the LASP1-∆LIM (aa 64–261) and LASP1-∆SH3 (aa 1–190) deletion mutants, we used the internal Ncol and Pst1 restriction sites in the human LASP1 cDNA, respectively. The fragments were recloned into pGEX4T1 [27].

Full-length cDNA of human LASP1 [28] was cloned into Venus-pcDNA3 (a kind gift from A. Miyawaki [29]) using the BamH1/EcoRI restriction sites.

The 3HA-tagged CXCR4 in pcDEF3 was kindly provided by M.J. Smit (Vrije University, Amsterdam, The Netherlands) and was used to create the 3HA-CXCR4-CFP construct by fusing enhanced cyan fluorescence protein (eCFP; BD Bioscience Clontech, Heidelberg, Germany) in-frame to the C-terminal end of the receptor following S352. The effusion was done by the standard PCR extension overlap technique, thereby deleting the C-terminal stop-codon of the receptor and the initial methionine codon of the fluorescent protein.

Recombinant human CXCL12 (Peprotech, 300-28A, Rocky Hill, NJ, USA) was maintained in imaging buffer supplemented with 0.1% BSA. All mutants were verified by sequencing.

### 2.2. Cell Culture and Transfection

The human breast cancer cell lines MDAMB-231, BT-20 (triple-negative breast cancer metastasis), and MCF-7 (luminal breast cancer), as well as the chronic myeloid leukemia cell lines K562 and LAMA-84, were purchased from ATCC (Manassas, VA, USA) and grown in RPMI 1640-GlutaMAX™ (Gibco BRL, Wiesbaden, Germany). Human embryonic kidney 293 (HEK-293) cell line was purchased from ATCC (Manassas, VA, USA) and cultured in DMEM high glucose (Gibco BRL). All media were supplemented with 10% FBS (Gibco BRL) and Penicillin/Streptomycin (PAA Laboratories, Coelbe, Germany). All cells were cultivated in a 5% CO_2_ atmosphere at 37 °C in a fully humidified incubator.

The MDAMB-231 cell line, stably transfected with LASP1 specific shRNA, was generated earlier [3], while BT-20 and MCF-7 breast cancer cells were transfected with LASP1 specific siRNAs, as described before [30]. Knockout of LASP1 in K562 cells was performed before by CRISPR/Cas9 [10]. HEK-293 cells were transfected with empty pcDNA3 as a control or pcDNA3 containing the cDNA for human HA-AKT1 (AKT1-WT), HA-AKT1-aa1-408 (AKT1-ΔC), HA-AKT1-L420S, HA-AKT1-L425S, or HA-AKT1-R465S using 0.2 µg/mL DNA and 3 µL/mL FuGENE^®^ HD for 24 h following the manufacturer’s instructions (Promega, Mannheim, Germany).

Cells were lysed in RIPA+-buffer (20 mM Tris pH 7.4, 150 mM NaCl, 1% sodium-deoxycholate, 1% Triton-X-100, 0.1% SDS, 1 mM sodium orthovanadate, 10 mM sodium-pyrophosphate, 10 mM EDTA, and protease inhibitor mixture). Western immunoblot analysis was performed to assess equal expression.

### 2.3. Western Blot

Protein lysates were subjected to 10% SDS-PAGE and blotted onto nitrocellulose membrane (GE Healthcare, Freiburg, Germany). Equal amounts of cells (1 × 10^4^) were analyzed by immunoblotting with the following antibodies: LASP1 [28] (diluted 1:2000); anti-β-Actin (1:3000, #1616, Santa Cruz, Dallas, CA, USA); pLASP1-S146 (1:700, IgG-1445, Immunoglobe, Himmelstadt, Germany); pLASP1-Y171 (1:500, IgG-1418, Immunoglobe); AKT1 (1:100, SAB 4500802, Sigma-Aldrich, Darmstadt, Germany); pAKT-S473 (1:1000, #9271, Cell Signaling, Danvers, MA, USA); pAKT-T308 (1:1000, #9275, Cell Signaling), pERK 1:500, #9102, Cell Signaling), HA-tag (1:700, sc-7392, Santa Cruz, Dallas, TX, USA).

Before overnight incubation at 4 °C with primary antibodies, membranes were blocked with 3% dry milk (Biorad, Munich, Germany) in TBS-T (Tris-buffered saline containing 0.1% Tween 20) for 1 h at room temperature. Immunoblots were probed 1 h with a secondary goat-anti-rabbit or goat-anti-mouse horseradish peroxidase conjugated antibody purchased from Biorad (1:5000) and developed by using the enhanced chemiluminescence reagent (GE Healthcare, Freiburg, Germany). Chemiluminescence images were taken using the Amersham Imager 600(GE Healthcare, Freiburg, Germany) and quantified by the Image QuantTL software (GE Healthcare, Freiburg, Germany).

### 2.4. Generation of GST-LASP1, His-LASP1 and GST-CRKL

The generation and purification of GST-LASP1 [3], His-LASP1 [15], and GST-CRKL [11] labeled proteins has been described in detail earlier.

### 2.5. GST-LASP1 Phosphorylation

A total of 5 µg GST-LASP1 was phosphorylated for 30 min by 0.5 µg recombinant active Lyn kinase (Sigma-Aldrich) according to the supplier’s recommendations at 30 °C in 100 µL phosphorylation buffer (5 mM Tris, 1 mM MgCl_2_, 0.1 mM DTT, and 100 µM ATP). PKA phosphorylation was performed with 0.3 µg purified C-subunit [31] at 30 °C in a total volume of 100 µL containing 20 mM Tris/HCl buffer pH 7.4, 10 mM MgCl_2_, 5 mM β-mercaptoethanol (Gibco, BRL), 0.01% (*w*/*v*) bovine serum albumin (Figure 1B).

### 2.6. Immunoprecipitation and Pull-Down

LASP1 immunoprecipitation (IP) in K562 and MDAMB-231 cells was performed by rotation of 2 × 10^6^ cells in 500 µL RIPA+-buffer with 2 µg LASP1 monoclonal B8 antibody (Nanotools, Freiburg, Germany) for 2 h at 4 °C, followed by incubation with A/G sepharose beads (Santa Cruz) for another 2 h at 4 °C. After washing the sepharose beads two times with ice-cold PBS, AKT1 binding to LASP1 was analyzed by Western blot.

For LASP1 pull-down, 2 × 10^6^ HEK-293 cells in 500 µL RIPA+-buffer (substituted with 160 nM nilotinib to inhibit BCR-ABL activity and His-LASP1 tyrosine phosphorylation) were incubated with 20 µg His-tagged LASP1 for 2 h at 4 °C, followed by incubation with 50 µL Ni-NTA beads (GE Healthcare, Freiburg, Germany) for 60 min. The immune complex was washed twice with ice-cold PBS buffer and prepared for SDS-PAGE.

### 2.7. Generation of Cxcr4^−/−^ Bone Marrow Derived Dendritic Cells

Cxcr4-floxed mice [32], kindly provided by Dr. Zou (Columbia University, New York, NY, USA, C57Bl/6 background), were crossed with C57Bl/6 CD11c-cre^+^ mice to obtain *CD11c-cre^+^ Cxcr4^+/+^* (WT) and *CD11c-cre^+^ Cxcr4^flox/flox^* mice (*Cxcr4^−/−^*) mice. BM-derived dendritic cells were generated from these, as described previously [33]. Briefly, femurs were removed from donor mice, the bone marrow was flushed and 1 × 10^6^ cells were cultured in 24-well plates in RPMI-1640 with 2 mM L-Glutamine (Gibco BRL) containing 10% heat-inactivated fetal calf serum, 100 U/mL penicillin/streptomycin, and β-mercaptoethanol (50 μM, Gibco BRL) supplemented with 50 ng/mL GM-CSF (Peprotech). Medium was renewed at days 3 and 5. BMDCs were used at day 7.

### 2.8. Radioactive Kinase Assay

Constitutive active Lyn kinase and ABL kinase were purchased from Sigma-Aldrich and BPS Biosciences (San Diego, CA, USA). Assay was performed with 0.01 µM kinase and 0.1 µM substrate (His-LASP1 and GST-CRKL) in 50 mM Tris-HCl, pH 7.5, 150 mM NaCl, 0.25 mM DTT, 0.1 mM EGTA at 30 °C in the presence of 50 µM [^32^P]-ATP for the times indicated.

### 2.9. PCR-Primer for Chemokine Receptors

mCXCR4-fwd: 5′-CCATGGAACCGATCAGTGTG

mCXCR4-rev: 5′-TGAAGTAGATGGTGGGCAGG

mCXCR7-fwd: 5′-GAAGCCCTGAGGTCACTTGG

mCXCR7-rev: 5′-CACAGTGTCCACCACAATGC

### 2.10. Life Cell Imaging

Prior to transfection, HEK-293 cells were seeded on individual 24 mm glass coverslips placed in wells of a 6-well plate. The coverslips were pre-incubated with ploy-D-lysine for 30 min and washed once with PBS. After 6 h, the media was replaced with fresh media and the cells were transfected with 500 ng 3HA-CXCR4-CFP and 50 ng of Venus-LASP1 per well using the Effectene transfection reagent (Qiagen, Hilden, Germany) according to the manufacturer’s instructions. Cells were used for confocal imaging 48 h after transfection. For the measurements, coverslips were mounted on an Attofluor^TM^ holder and kept in 190 µL of imaging buffer (140 mM NaCl, 5.4 mM KCl, 2 mM CaCl2, 1 mM MgCl2, 10 mM HEPES; pH 7.3) containing 0.1% BSA. Then, 10 µL of 20 µM CXCL12 were carefully added into the solution to reach a final ligand concentration of 1 µM. The recording started at the same time as the ligand was added.

The experiments were performed using a Leica TCS SP8 confocal microscope (Leica, Wetzlar, Germany). Images were taken using a 63× water objective (numerical aperture, 1.4). CFP was excited at 442 nm using a diode laser. CFP emission fluorescence intensity was detected from 455 to 510 nm. The YFP was excited with an argon laser at 514 nm and the emission fluorescence intensity was detected from 525 to 600 nm. The time series was recorded using sequential excitation, and images were taken at 3 s intervals for 20 min with a 512 × 512-pixel format, line average 1, frame average 1, and 400 Hz.

### 2.11. Immunofluorescence

For immunofluorescence microscopy, 3 × 10^4^ K562 cells were fixed by adding 10% formalin into the culture medium for a final concentration of 5%. After 10 min incubation at 4 °C, cells were washed twice with cold PBS at 1000× *g* for 2 min at 4 °C, followed by centrifugation for 10 min at 140× *g* at RT to generate cytospins [11]. MDAMB-231 cells were grown until 70% confluence on glass coverslips and fixed in 4% (*w*/*v*) paraformaldehyde in PBS on ice. Fixed cells were permeabilized with 0.1% (*w*/*v*) Triton X-100 in PBS and stained with monoclonal AKT1-Alexa Fluor 488 (Merck, Darmstadt, Germany #16-293, 1:100 dilution) and affinity-purified polyclonal pLASP1-S146 PE-labelled antibody (diluted 1 µg/mL), as well as CXCR4-Alexa Fluor-488 antibody (Abcam #208128, diluted 1:250). PE-labeling of rabbit polyclonal pLASP1-S146 was performed with the Abcam R-Phycoerythrin conjugation kit (#102918, Abcam, Cambridge, UK), following the manufacturer’s instruction. Omnifocal analysis was done with a Biorevo BZ-9000 (Keyence, Neuisenburg, Germany).

### 2.12. SH3 Domain Mapping

The 3D protein structure of the LASP1-SH3 domain was designed using PyMOL Molecular Graphics System, Version 2.0.4 (Schrödinger, LLC, New York, NY, USA) open-source software.

### 2.13. Statistics

Paired two-tailed Student’s t-test for dependent groups was applied to determine the effects of LASP1 knockdown on AKT1-S473 phosphorylation. Results were considered significant at *p* < 0.05 (* *p* < 0.05; ** *p* < 0.01; *** *p* < 0.001; n.s. not significant).

## 3. Results

### 3.1. LASP1 Depletion Negatively Regulates AKT1-S473 Phosphorylation

To confirm previous observations of the effect of LASP1 knockdown on AKT1 phosphorylation [22,24,34], we performed experiments with MDAMB-231 and MCF-7 breast cancer cells, stably transfected with doxycycline-inducible LASP1-specific shRNA, transiently transfected with different LASP1-specific siRNAs. After two days in cell culture, cells were tested for LASP1 depletion and AKT1-S473, AKT1-T308, and ERK phosphorylation by immunoblotting (Figure 1A). Data revealed a significant reduction in S473 phosphorylation (Figure 1B) after LASP1 depletion in both cell lines, whereas T308 and ERK1/2 phosphorylation were not affected (Figure 1A). The antibiotic doxycycline itself had no effects on AKT1 phosphorylatiosr results were obtained with different LASP1-depleted K562 single clone cell lines (Figure 1).

### 3.2. LASP1-AKT1 Interaction in MDAMB-231 Breast Cancer Cells

Based on the previous data showing a LASP1-mediated PI3K/AKT1 activation [22,24,34] and our own findings demonstrating overexpression of LASP1 in breast cancer cells [35] and in CML [11], we wanted to address whether there is a direct regulation of AKT1 by LASP1 in these cells. We therefore performed an immunoprecipitation assay with monoclonal LASP1 antibody and indeed could detect an LASP1-AKT1 interaction in MDAMB-231 breast cancer cells but not in K562 cells. However, when performing a pulldown assay with His-LASP1 in K562 cell lysates, AKT1 binding to LASP1 was also detectable (Figure 2A).

To test for differences in LASP1 in MDAMB-231 and K562 cells, we measured the phosphorylation status of the protein in both cell lines. Interestingly, K562 cells mainly showed LASP1-Y171 phosphorylation, while MDAMB-231 cells demonstrated both phosphorylation at Y171 and S146 (Figure 2B). This difference in phosphorylation pattern was also seen in MCF-7 breast cancer cells and LAMA-84 CML cells (Figure 2B). When calculating a ratio between pLASP-S146 and pLASP-Y171, data revealed a 2-3-fold higher Y171 phosphorylation in CML cells compared to breast cancer cells. However, be aware that the S146 and Y171 phosphorylation intensities do not reflect absolute phosphorylation due to different antibody specificities.

### 3.3. AKT1 Preferentially Binds to LASP1 Phosphorylated at S146

These differences in LASP1 phosphorylation prompted us to test LASP1-AKT1 interaction with respect to different LASP1 phosphorylation states. Pulldown assays were performed with recombinant His-LASP1, either phosphorylated at S146 by constitutive active C-subunit of PKA, or at Y171 by active Lyn kinase, and lysates of HEK-293 cells transiently transfected with HA-tagged full-length AKT1. As depicted in Figure 2C (and Figure 3E), AKT1 bound to unphosphorylated LASP1, but binding to pLASP1-S146 was increased, whereas phosphorylation at Y171 impaired AKT1 binding. Phosphorylation was controlled by specific antibodies generated against the two LASP1 phosphorylation sites (Figure 2D).

### 3.4. Interaction between Linker Region of LASP1 and AKT1 C-Terminus

It is known that the SH3 domain modulates a negatively charged cleft by aromatic amino acids, which interact with proline-rich sequences flanked by positively charged residues (x^+^) like x^+^xPxxP or xPxxPx^+^ [36]. We used the open source molecular graphic software PyMOL™ to modulate the SH3 domain of LASP1. In Figure 3A, the domain is shown in a Coulombic surface presentation. The conserved aromatic amino acids are depicted as stick models. Concurrently, we identified two putative ligand sequence PxxP motifs with dedicated proline-flanking, negatively-charged amino acids as consensus target site for SH3 binding at the C-terminus of AKT1, near the S473 phosphorylation site.

To assess a possible interaction of the LASP1-SH3 domain with the C-terminus of AKT1, we mutated K420, K426, and R465 (marked in red, Figure 3B) of the full-length AKT (AKT1-WT) to serine. In addition, we tested truncated AKT1-ΔC (aa 1–418) and deletion mutants of LASP1-ΔSH3 (aa 1–190) and -ΔLIM (aa 64–261) to map the defined binding region on LASP1 to AKT1. HEK-293 cells were used as an exogenous system because of high transfection efficiency.

Pulldown experiments with GST-LASP1-WT and endogenous AKT1-WT as well as AKT1-ΔC confirmed the suggested interaction between LASP1 and the C-terminus of AKT1, as truncated AKT1 failed to bind to LASP1 (Figure 3C,D). Unexpectedly, LASP1-ΔSH3 showed no reduction in binding to AKT1-WT (Figure 3E). These data were further confirmed by the AKT1-mutants K420S, K426S, and R465S, as the loss of any of these positively charged amino acids had no negative effect on LASP1 binding (Figure 3D). Binding was also detected between AKT1 and LASP1-ΔLIM. By deductive analysis, the AKT1-C-terminus favors binding to the S146-phosphorylated LASP1 nebulin-linker region.

Immunofluorescence staining of MDAMB-231 cells confirmed the co-localization of LASP1 with AKT1 at the cell membrane (Figure 4A, white arrows) and around the nucleus (Figure 4A, orange arrow), most likely at endosomes. In addition, AKT1 expression was verified in the nucleus. A more detailed analysis using phycoerythrin (PE)-labeled pLASP-S146 antibody showed the pLASP-S146-AKT1 interaction only around the nucleus (Figure 4B, orange arrows) but not along the membrane. In CML cells, immunofluorescence detected an exclusive cytosolic expression of LASP1, while AKT1 is present in the cytosol and in the nucleus (Figure 4C). The observed co-localization in the cytosol is most likely incidental because of the large nuclei and the reduced cytoplasm volume. Staining of CML cells with PE-labeled pLASP-S146 was negative and affirmed (a) the lack of this LASP1 isoform in CML (as already observed in Western blots; Figure 2B), and (b) the specificity of the labeled antibody (data not shown).

### 3.5. CXCR4-LASP1 Binding

Binding of LASP1 to the C-terminus of CXCR4 is dependent on LASP1-S146 phosphorylation [6]. Since in CML LASP1 is highly phosphorylated at Y171 (Figure 2B), we wondered how this phosphorylation influences LASP1 binding to the receptor. We performed co-immunoprecipitation experiments with phosphorylated His-LASP1 and the recombinant C-terminus of CXCR4 bound to GST. As seen in Figure 5A, binding to the CXCR4 C-terminus only occurred with pLASP1-S146 but not pLASP1-Y171 or unphosphorylated LASP1.

Subsequently, we investigated whether phosphorylation of LASP1 at one site may interfere with phosphorylation at another site. Therefore, purified recombinant GST-LASP1, bound to Sepharose beads, was first phosphorylated by PKA or Lyn kinase, kinases were then washed off, and beads were resuspended in new buffer containing the opposing kinase. As shown in Figure 5B, phosphorylation of LASP1 at one site hinders phosphorylation at the second site. This suggests either conformational changes by phosphorylation in the protein [37] or a charge repulsion between the kinase and the additional negative phosphate group, as positive-charged amino acids (PKA) or hydrophobic residues (Lyn kinase) are required for optimal substrate binding [38].

### 3.6. CXCR4-Induces LASP1 Phosphorylation in Intact Cells

Given the fact that CXCR4 is known to function as an upstream regulator of AKT1, combined with identification of LASP1 as a phosphorylation-dependent binding partner of CXCR4 and AKT1, we addressed if there was a “Yin-Yang” balance between serine and tyrosine phosphorylation of LASP1 and whether the above-mentioned in vitro observations were of any relevance *in vivo*? To investigate this question we used murine bone marrow-derived dendritic cells (BMDCs), as these cells harbor the chemokine receptor CXCR4 and share some common signaling pathways with leukemia stem cells [39]. Subsequently, the Cre/loxP system was used to generate control cells with deleted CXCR4. qRT-PCR analyses confirmed the deletion of CXCR4 in CD11c-cre^+^
*Cxcr4*^flox/flox^ compared to CD11c-cre^+^
*Cxcr4*^+/+^ mice (data not shown). After stimulation with the CXCR4 agonist CXCL12, cells were analyzed for AKT1 and LASP1 phosphorylation by Western blotting.

In a time-resolved analysis, a three-fold AKT1-S473 phosphorylation was observed within 2 min of CXCL12 stimulation, which declined back to basal levels after 10 min in control cells (Figure 6B, green line). Simultaneously, cells revealed a 50% increase in LASP1 phosphorylation at Y171 (Figure 6B, red line), and depicted a dephosphorylation at T156 (corresponding to S146 in human LASP1 [40] and recognized by the pLASP1-S146 antibody due to high sequence homology (Appendix A)), which reverted back to basal levels within 10 min (Figure 6B, blue line). In the CXCR4-deficient cells, basal AKT1 phosphorylation levels diminished, and CXCL12 stimulation triggered a reduced pAKT1-S473 phosphorylation compared to controls cells (Figure 6B, black line). Basal LASP1 phosphorylation was strongly reduced by CXCR4 deficiency, and no LASP1 phosphorylation could be induced by CXCL12 (Figure 6B, grey line), suggesting CXCR4 being a major signaling receptor for LASP1 activation.

It should be noted that we also checked for CXCR7 levels, as CXCL12 can also partially act through CXCR7 [41]. However, no expression was detected in the cells by qRT-PCR (data not shown).

### 3.7. LASP1-CXCR4 Co-Localization in Intact Cells by Life Cell Imaging

To further analyze the LASP1-CXCR4 interaction, we attempted to visualize the co-localization of both proteins in MDAMB-231 cells. However, immunofluorescence staining of CXCR4 failed due to low receptor expression and unspecific antibody recognition. It should be noted that we tested four different commercially available CXCR4 antibodies (antikoerper-online, sc-9046, sc-53534, and abcam UMB2), among which only one showed a positive protein product of the correct molecular weight (not shown). We therefore generated CXCR4-CFP and Venus-LASP1 and transiently expressed the constructs in HEK-293 cells. Under basal conditions, a co-localization of both proteins at the cell membrane was seen (Figure 7, white arrows at 0 min). Additional cytosolic LASP1 expression was observed. Stimulation of the cells with 1 µM CXCL12 revealed an internalization of the CXCR4 receptor over 8 min (Figure 7 and Appendix A). In parallel, LASP1 binding to CXCR4 declined during the first 2–3 min but increased again after 7–8 min, along with the internalized receptor (Figure 7, orange arrows). Western blot analysis of this experiment monitored increased AKT1 activation (phosphorylation of AKT1 at S473) in the presence of CXCR4, concomitant with partial pLASP1-Y171 phosphorylation/pLASP1-S146 dephosphorylation during the first minutes, and followed by a return to initial phosphorylation levels after 10 min (Figure 8). This paralleled the phosphorylation–dephosphorylation time-schedule observed before in dendritic cells (Figure 7).

### 3.8. Enhanced pLASP1-S146 in K562 CML Cells during TKI Treatment

To substantiate our observations concerning the phospho-serine/phospho-tyrosine cross-regulation, we set out to test LASP1 phosphorylation in K562 cells before and after the inhibition of constitutively active tyrosine kinase BCR-ABL. The phosphorylation status of the adaptor protein CRKL (Figure 9A, DMSO control) was used to monitor the efficacy of tyrosine kinase inhibitor (TKI) treatment [42]. After 24 h incubation with 60 nM and 120 nM nilotinib, phosphorylation of LASP1 at S146 increased, whereas CRKL showed a decrease in phosphorylation (Figure 9A). However, high LASP1-Y171 phosphorylation levels remained unaltered (Figure 9A), possibly due to phosphorylation by other active tyrosine kinases in leukemia cells that were not inhibited by nilotinib [43]. Only after treatment with the dual ABL/Src TK inhibitor dasatinib, a partial dephosphorylation of pLASP1-Y171, was observed (Figure 9A); phosphorylation of CRKL was completely blocked under these conditions, assuming different substrate specificities for the tyrosine kinases in CML. To verify these observations, we performed radioactive kinase assays with constitutive active ABL and Lyn kinase and LASP1, CRKL as substrates. Graphic visualization of the analyzed radioactive gels revealed faster and stronger phosphorylation of LASP1 by Lyn compared to ABL kinase, whereas CRKL seemed to be a much less preferred substrate of both kinases (Figure 9B). These findings suggest that in CML patients, the preferential substrate recognition of LASP1 by Lyn kinase could serve as a new read-out in patients with beginning TKI resistance, given that these patients show higher Lyn kinase activity as a characteristic of emerging BCR-ABL-independent resistance [44,45].

## 4. Discussion

AKT1 function is either regulated by direct inhibition/stimulation or by affecting subcellular localization. Activation mediates downstream responses, including cell survival, growth, proliferation, cell migration, and angiogenesis [20]. This study as well as several other publications demonstrated that LASP1 knockdown decreased AKT-S473 phosphorylation [21,22,23,24,34]. Concordantly, LASP1 knockdown inhibited cell migration and proliferation in several tumor entities [2] and an indirect regulation of AKT1 by LASP1 was suggested. However, the authors had not taken into account the different phosphorylation statuses of LASP1 when performing immunoprecipitation experiments or pull-down assays to identify possible LASP1 binding partner. The study by Goa et al. convincingly demonstrated an increased activation of AKT1, at least in part, by LASP1 targeting PTEN and inactivating PI3K [23].

In the current study, we analyzed the newly identified direct phosphorylation-dependent binding of LASP1 to AKT1 and CXCR4, two major players in tumor progression. In view of the new results and in combination with earlier published data, we propose a model that explains the divergent roles of LASP1 and CXCR4 in solid breast cancer and in CML cells (Figure 10).

In breast cancer, LASP1 is partially phosphorylated at S146 by PKA [3] or at Y171 by not otherwise specified phosphotyrosine kinases (PTKs) (Figure 2B). Phosphorylation at S146 enhances binding to the C-terminus of CXCR4 (Figure 5A), a chemokine receptor overexpressed on malignant cells and associated with EMT, upregulation of chemokine receptors, and the secretion of cytokines (IL-6, MCP-1, and GM-CSF), thereby enhancing migration, lymphatic invasion, and metastatic activity [17,46]. The interaction occurs at the LKIL motif (AA 327–330) [6], a region critical for proper receptor internalization and degradation [47].

Based on our studies, we suggest that binding of pLASP1-S146 at the C-tail of CXCR4 stabilizes the receptor and sterically hinders the phosphorylation at multiple PKC serine phosphorylation sites important for receptor internalization, deactivation, and recycling back to the plasma membrane [47].

Upon stimulation by CXCL12, heterotrimeric G-protein is activated. Gα_i_ inhibits adenylyl cyclase/cAMP/PKA signaling and activates Src family tyrosine kinases, as seen in the transient dephosphorylation of LASP1 at the PKA-site (S146) and concomitant phosphorylation at Y171 (Figure 6 and Figure 8). Phosphorylation of LASP1 at Y171 by Src and ABL kinase has already been observed earlier [4,11]. Increased LASP1-Y171 phosphorylation renders the protein less affine to the CXCR4 C-terminus (Figure 5A), hence, the facilitation of serine and tyrosine phosphorylation of the receptor by repelling/unblocking [47].

Concomitantly, Gβγ release results in the activation of several pathways, i.e., PI3K/AKT [48], a key player in tumor cell survival and proliferation [19]. The new data presented in this paper now demonstrate an additional interaction of LASP1 and pLASP1-S146 with the C-terminus of AKT1 (Figure 3E). A straightforward scaffold model predicted the dependence of AKT1 interaction with LASP1 at the cell membrane, potentially in combination with mTORC2, as AKT1 phosphorylation at T308 by PDK1 was not affected (Figure 1). Indeed, the presented immunofluorescence staining demonstrates a co-localization of LASP1 and AKT1 at the cell membrane, as well as an interaction of pLASP1-S146 and AKT1 around the nucleus (Figure 4). So far, we cannot rule out LASP1-AKT1 co-localization around the nucleus as well. However, as already shown before, only phosphorylation of LASP1 at S146 reduces protein binding to F-actin and induces translocation from the membrane to the cytoplasm [28] as well as into the nucleus when bound to the shuttle protein zona occludens 2 [3]. In the nucleus, LASP1 stabilizes Snail, a transcriptional repressor of adherens junction proteins [14], and enables matrix metalloprotease expression by modulating AP-1 transcriptional activity [15], processes that are involved in mesenchymal cell formation and tumor cell migration.

In contrast to breast cancer cells and other solid tumors [2], LASP1 is not detected in the nucleus of CML cells, either in its tyrosine-phosphorylated form or after serine phosphorylation by forskolin-stimulated PKA activation [11]. Therefore, a direct transcriptional impact of LASP1 in CML is virtually excluded.

In CML, chromosomal translocation results in an oncogenic BCR-ABL gene fusion that encodes for the constitutively active BCR-ABL fusion protein. As a result, hyper-phosphorylation of proteins is observed, among them the newly identified BCR-ABL substrate LASP1 [11]. In contrast to MDAMB-231 breast cancer cells, K562 CML cells show almost exclusively pLASP1-Y171 phosphorylation (Figure 2B), the CXCR4 receptor is down-regulated [8], and no binding or receptor stabilization is expected. Likewise, the oncogenic AKT1 activity is not attenuated by pLASP1-Y171. As described in the literature, treatment of CML cells by TKI up-regulates CXCR4, thus promoting cell migration back to bone marrow stroma [8,49], and therefore results in BCR-ABL inhibition, decreased tyrosine phosphorylation, and restored PKA activation [50]. This was partially confirmed in our K562-nilotinib experiments, showing decreased pCRKL-Y207 phosphorylation, concomitant with increased pLASP-S146 phosphorylation (Figure 9A), enabling binding to and stabilization of CXCR4. In K562 cells, pLASP1-Y171 dephosphorylation occurred only after dasatinib treatment, as LASP1 is also a preferential substrate for other tyrosine kinases (Figure 9B). In patients, an impaired LASP1-Y171 phosphorylation was seen under TKI therapy [11], but unfortunately, serine phosphorylation was not checked in that study. However, a recent publication showed imatinib-induced vasodilator-stimulated phosphoprotein (VASP) phosphorylation on S157 in K562 cells and in CML patients’ bone marrow cells after imatinib treatment [51]. Completive data revealed a VASP-Y39 phosphorylation in BCR-ABL-positive leukemic cells [52].

In conclusion, we postulate a model for the divergent CXCR4-LASP1-AKT1 signaling in breast cancer and CML by differences in LASP1 phosphorylation. While pLASP1-S146 may contribute to CXCR4 stabilization in solid tumors, thereby promoting metastasis, LASP1-Y171 phosphorylation by BCR-ABL obviously hinders LASP1-CXCR4 binding and renders the chemokine receptor more sensitive to degradation in leukemic cells, thus favoring mobilization (Figure 10). Therefore, surveillance of LASP1 phosphorylation status in CML patients may serve as a new read-out in CML patients under TKI treatment; appropriate studies are underway.

## Figures and Tables

**Figure 1 cells-09-00444-f001:**
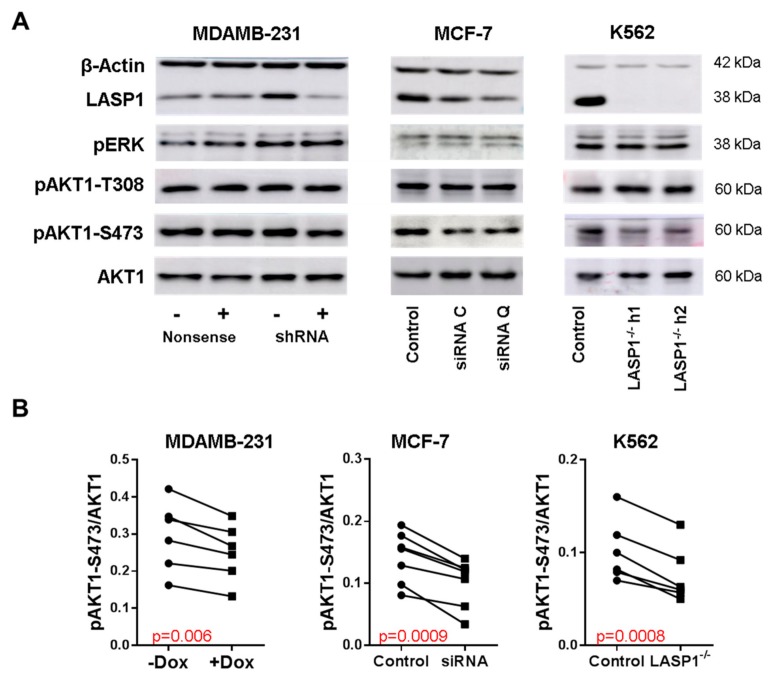
LASP1 regulates AKT1-S473 phosphorylation. (**A**) Representative Western blots analyzing LASP1, pERK, AKT1, pAKT1-T308 and pAKT1-S473 after LASP1 knockdown in MDAMB-231 (stably transfected with shRNA-LASP1 or nonsense shRNA and downregulated by inducible doxycyclin treatment (depicted with (+) and (−))), in MCF-7 breast cancer cells (transiently transfected with different LASP1 specific siRNA), and in LASP1-knockout K562 single cell lines. (**B**) Impaired AKT1-S473 phosphorylation after LASP1 depletion. Statistical differences were analyzed using the paired t-test (*n* = 6).

**Figure 2 cells-09-00444-f002:**
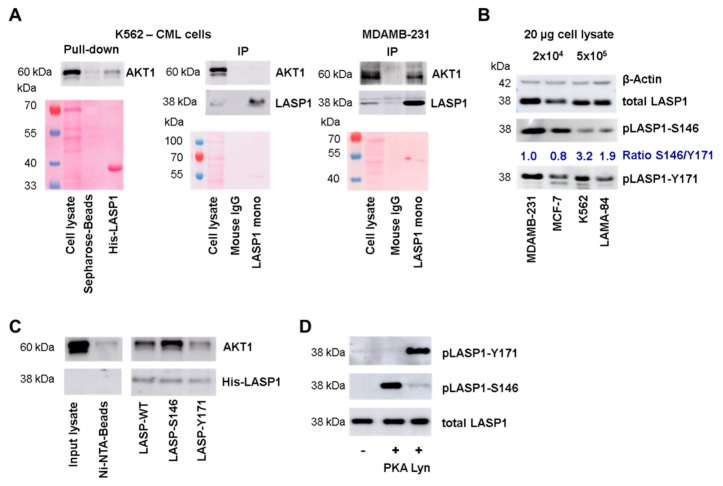
LASP1-AKT1 interaction. (**A**) Western blot analysis of AKT1 pulled-down by His-LASP1 from K562 cell lysate (left panel), or immunoprecipitated with LASP1 monoclonal B8 antibody from K562 (middle panel) or MDAMB-231 cell lysate (right panel). AKT1 immunoprecipitation is only seen in MDAMB-231 lysate. Ponceau staining revealed equal protein input. Experiments were repeated three times with similar results. (**B**) Western blot analysis of LASP1, pLASP1-Y171, and pLASP1-S146 in MDAMB-231, MCF-7, K562, and LAMA-84 cells. In chronic myeloid leukemia (CML) cells LASP1 is mainly phosphorylated at Y171, whereas MDAMB-231 and MCF-7 show LASP1 phosphorylation at Y171 and S146. β-Actin served as loading control. Comparable results were obtained in three experiments with different passages. (**C**) Western blot analysis of pulled-down AKT1 using His-LASP1 wild type (WT), LASP1 pre-phosphorylated at Y171 by active Lyn kinase or at S146 by constitutive active C-subunit of protein kinase A (PKA). AKT1 preferentially binds to pLASP1-S146. The experiment was repeated three times with equal results. (**D**) Western blot control of LASP1 pre-phosphorylation by active Lyn kinase at Y171 or by constitutive active C-subunit of PKA at S146.

**Figure 3 cells-09-00444-f003:**
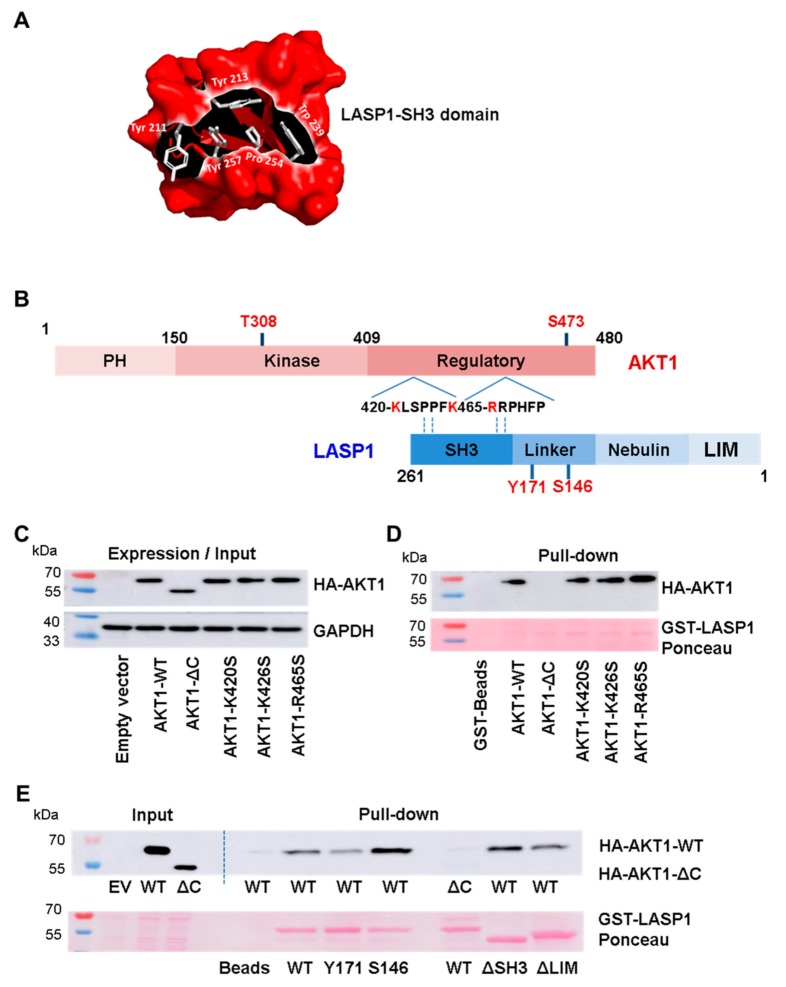
LASP1-AKT1 interaction (**A**) Coulombic surface presentation of LASP1-SH3 domain. Conserved amino acids are depicted as stick models. (**B**) xPxxPx sequence motifs within the proline-rich C-terminal regulatory AKT1 domain; negatively-charged amino acids, possibly interacting with the LASP1-SH3 domain, are marked in red. (**C**) Western blot input control of AKT1 expression in HEK-231 cells transfected with plasmids harboring AKT1-WT (aa 1–480), AKT1-ΔC (aa 1–408), AKT1-K420S, AKT1-K426S, and AKT1-R465S. GAPDH served as loading control. (**D**) Western blot analysis of AKT1 pull-down by GST-LASP1-WT from HEK-231 lysate. Data confirmed an interaction between LASP1 and the C-terminus of AKT1. (**E**) Western blot analysis of AKT1 pull-down by GST-LASP1-WT, GST-pLASP1-Y171, GST-pLASP1-S146, GST-LASP1-ΔSH3, and GST-LASP1-ΔLIM. Data confirmed impaired AKT1-ΔC binding to LASP1-WT. Ponceau-staining confirmed equal protein input; EV: empty vector; WT: wild type. All results were confirmed in three independent experiments.

**Figure 4 cells-09-00444-f004:**
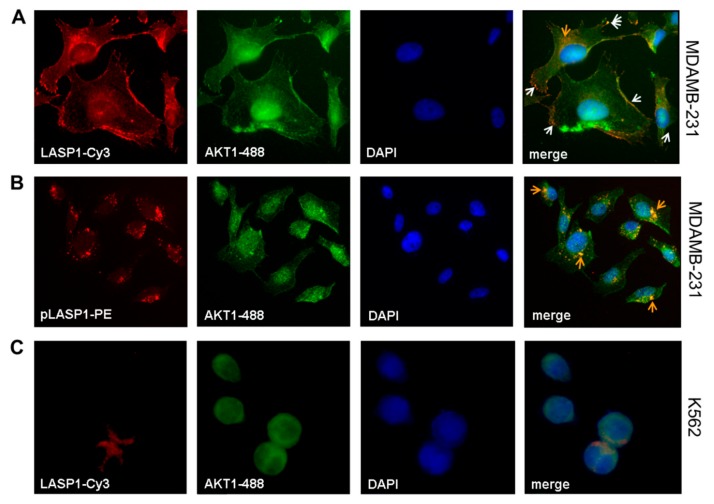
Co-localization of LASP1-AKT1 in MDAMB-231 but not in K562 cells. Merged immunofluorescences of (**A**) LASP1 (red, Cy3-labeled) and (**B**) pLASP1-S146 (red, PE-labeled) with AKT1 (green, 488-labeled) in MDAMB-231 cells revealed co-localization of LASP1 with AKT1 at the membrane (white arrows), while pLASP1-S146 is preferentially co-localized with AKT1 perinuclear (orange arrows). DAPI staining visualized the nuclei. Omnifocal images were captured at 60× magnification. In K562 cells (**C**), merged immunofluorescences of LASP1 (red, Cy3-labeled) with AKT1 (green, 488-labeled) showed coincidental cytosolic localization of both proteins. DAPI staining visualized the nuclei. Omnifocal images were captured at 100× magnification.

**Figure 5 cells-09-00444-f005:**
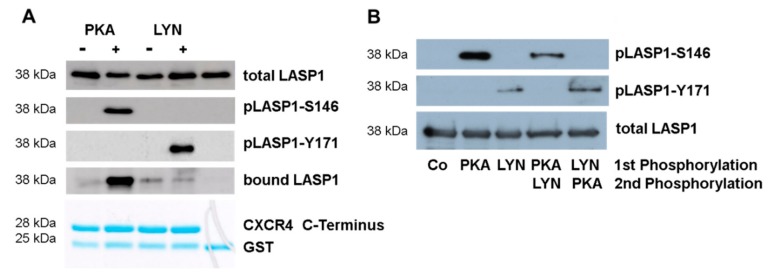
Binding of LASP1 to chemokine receptor 4 (CXCR4) is dependent on LASP1-S146 phosphorylation. (**A**) Western blot analysis of LASP1, pLASP1-Y171, and pLASP1-S146, co-immunoprecipitated with GST-CXCR4. Data revealed only binding of pLASP1-S146 with CXCR4. (**B**) GST-LASP1 beads were pre-phosphorylated by either constitutive active C-subunit of PKA or active Lyn kinase for 30 min, followed by washing and continuous incubation with the opposing kinase. Data indicate that phosphorylation of LASP1 at one site hinders phosphorylation at the second site. Experiments were repeated twice with similar results.

**Figure 6 cells-09-00444-f006:**
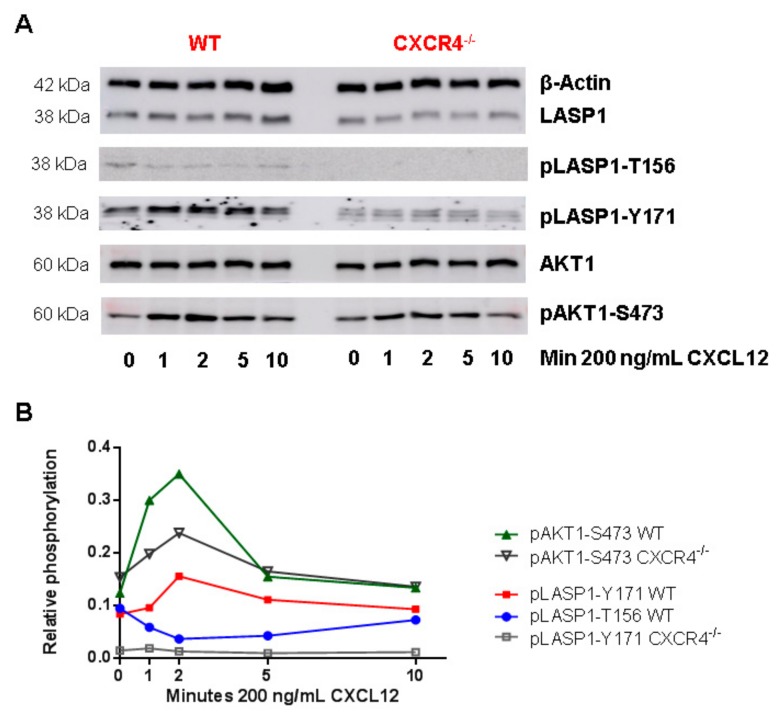
Role of CXCR4 stimulation on LASP1 phosphorylation in mouse dendritic cells. (**A**) Representative Western blot analysis of LASP1, pLASP1-Y171, pLASP1-S146, AKT1 and pAKT1-S473 in mouse dendritic cells harboring CXCR4-WT (*CD11c-cre^+^* Cxcr4^+/+^ mice (*wt*) or biallelic CXCR4 knockout (*CD11c-cre^+^ Cxcr4^flox/flox^*
*mice (Cxcr4^−/−^);* confirmed by qRT-PCR, data not shown) after stimulation with the CXCR4 agonist CXCL12 for the time points indicated. β-Actin served as loading control. (**B**) Densitometric quantification of phosphorylation. Data revealed a three-fold increase in pAKT1-S473 and a 1.5-fold increase in pLASP1-Y171 phosphorylation in WT cells compared to CXCR4 knockout cells after 2 min CXCL12 stimulation, which declined to basal levels after 10 min. Simultaneously, LASP1 was dephosphorylated at S146. Comparable results were obtained in four independent experiments.

**Figure 7 cells-09-00444-f007:**
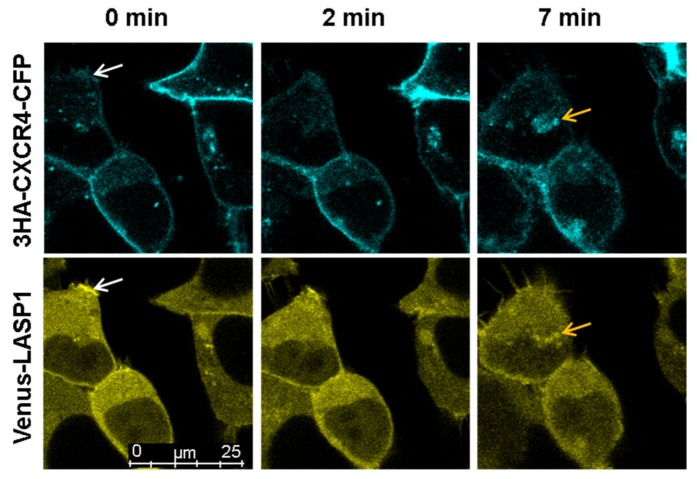
Life cell imaging of CXCR4 and LASP1 internalization after CXCL12 stimulation. HEK-293 cells were co-transfected with CXCR4-CFP and Venus-LASP1 for 48 h followed by 20 min stimulation with 1 µM CXCL12. A time series with images taken every 3 s was recorded using sequential excitation. Images representing time points 0 min (basal phosphorylation—LASP1 and CXCR4 are co-localized at the membrane, white arrows), 2 min (maximal dephosphorylation of pLASP1-S146—internalization of CXCR4), and 7 min (perinuclear co-localization of CXCR4 and LASP1, orange arrows) is shown. Images were captured at 63× magnification. The corresponding video is deposited in the Appendix A.

**Figure 8 cells-09-00444-f008:**
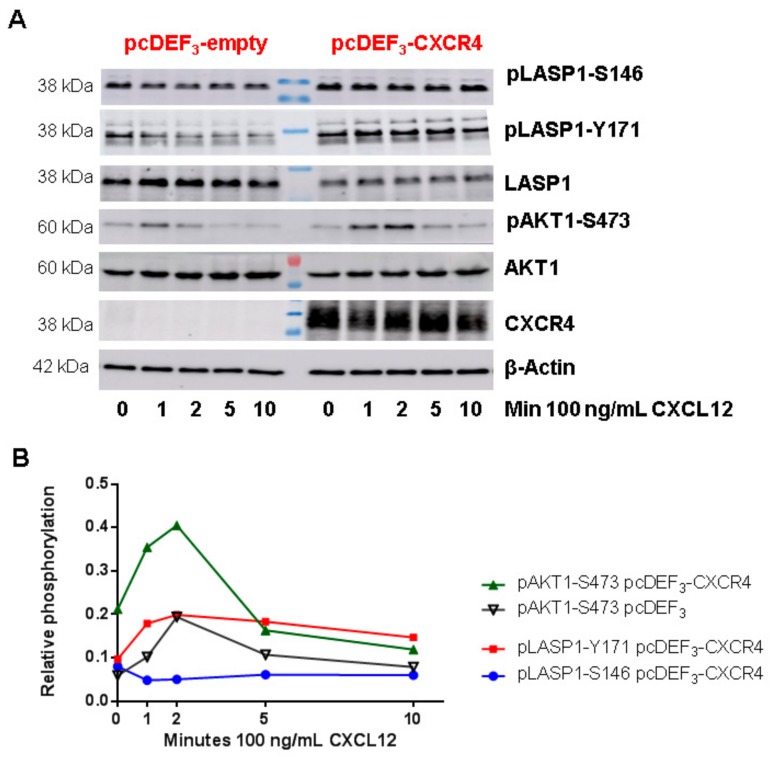
Role of CXCR4 stimulation in LASP1 phosphorylation in CXCR4 expressing HEK-293 cells. (**A**) Representative Western blot analysis of LASP1, pLASP1-Y171, pLASP1-S146, AKT1, and pAKT-S473 in HEK-293 cells transfected with CXCR4 and stimulation with the CXCR4 agonist CXCL12 for the time points indicated. β-Actin served as loading control. (**B**) Densitometric quantification of phosphorylation. Data revealed a two-fold increase in pAKT1-S473 and a two-fold increase in pLASP-Y171 phosphorylation concomitant with a 50% decrease in LASP1-S146 phosphorylation in CXCR4 expressing HEK-293 cells compared to CXCR4-deficient HEK-293 cells after 2 min CXCL12 stimulation, which declined to basal levels after 10 min. Comparable results were obtained in two independent experiments.

**Figure 9 cells-09-00444-f009:**
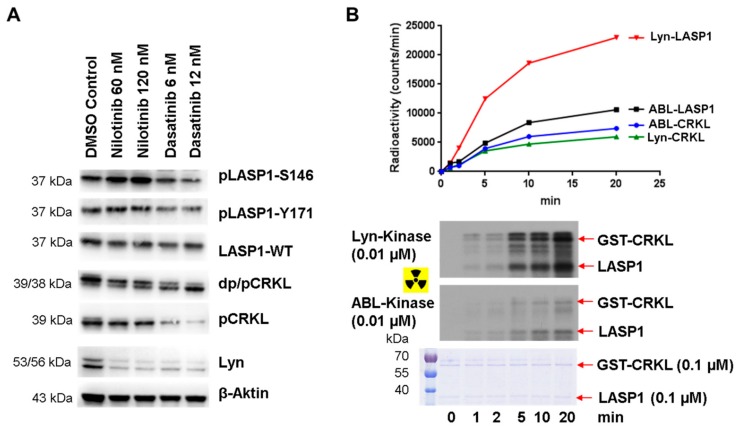
Differences in LASP1 phosphorylation in nilotinib and dasatinib treated K562. (**A**) Western blot analysis of K562 cells treated with nilotinib or dasatinib for 24 h showed different phosphorylation and dephosphorylation for LASP1 and CRKL. (**B**) Radioactive kinase assay demonstrated higher substrate affinity of Lyn kinase for LASP1 compared to CKRL.

**Figure 10 cells-09-00444-f010:**
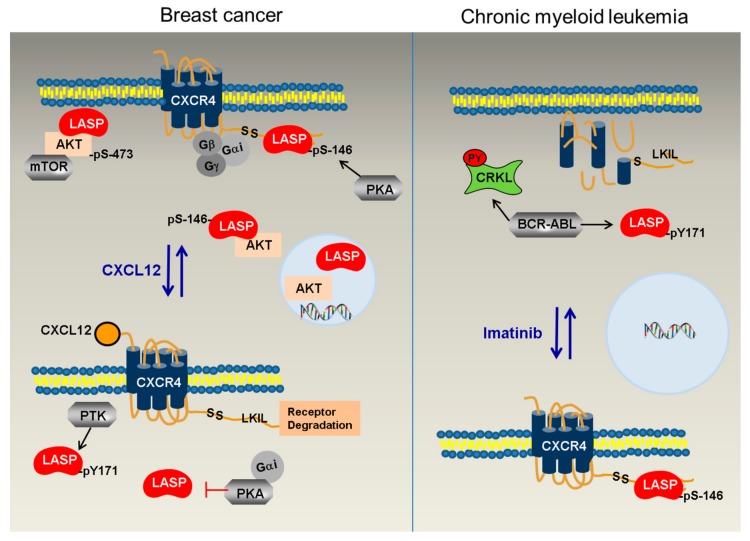
Postulated differences in the CXCR4-LASP1-AKT1 signaling between breast cancer and CML. In solid breast cancer (left panel), PKA is over-expressed and active, leading to phosphorylation of LASP1 at S146 and binding of the protein to CXCR4. In addition, LASP1 binds to AKT1 and, as a scaffolding protein, allows correct binding of AKT1 to the membrane (and potentially the mTORC2 complex) for efficient S473 phosphorylation. A quantity of AKT1-pLASP1-S146 is present perinuclear. Low levels of AKT1 and LASP1 are also detected in the nucleus to favor transcriptional activity. Stimulation of CXCR4 by CXCL12 leads to activation of downstream phosphotyrosine kinases (PTKs) and increases LASP1-Y171 phosphorylation. Simultaneously, released Gαi inhibits the PKA pathway and LASP1 becomes dephosphorylated at S146, detaches from the receptor tail, and renders CXCR4 more sensitive for degradation. In CML (right panel), constitutively active tyrosine kinase BCR-ABL results in predominant pLASP-Y171 phosphorylation, impeded CRKL-LASP1 binding, and down-regulation/degradation of CXCR4 receptor. No LASP1 is detected in the expanded nucleus. After inhibition of BCR-ABL by TKI like nilotinib, CXCR4 is restored in the plasma membrane, LASP1-Y171 phosphorylation decreases, while pLASP1-S146 increases and allows binding to and stabilizing of the chemokine receptor for homing.

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
