# Peer review of "Phosphorylation-Dependent Differences in CXCR4-LASP1-AKT1 Interaction between Breast Cancer and Chronic Myeloid Leukemia"

_cells, 2020, doi:10.3390/cells9020444_

Round 1
Reviewer 1 Report
The Manuscript of Butt et al. presents an interesting study describing a novel direct binding of LASP1 to both CXCR4 and Akt1, showing a different phosphorylation state comparing breast cancer and CML cell lines, where CXCR4 plays a quite different role.
All in all, the study is of interesting for the field, reporting a novel LASP1-Akt1 interaction and the comparison of the mechanism in two different tumor entities where CXCR4 has a different role is attractive. However, the quality should be improved adding additional experiments and corrections to make this manuscript suitable for publication in Cells.
Main points:
Why the authors use only these two specific cell lines (MB231 and K562)?
Since the authors claim that this mechanism is generic for BC (breast cancer) and CML, they should use more cells lines to prove that, and not only one for each tumor entity (i.e. at least the first data, Fig.1 and 2, should be performed using several cell lines for both BC and CML to corroborate this statement).
Moreover MDA-MB-231 is a triple negative breast cancer cell line, is this is true for different breast cancer subtype (i.e. luminal)?
Fig.1: This WB analysis should be performed also in K562, since the authors are comparing these two different cell lines.
Fig.1A: Authors should explain why the expression level of LASP1 in all shRNAs – Dox are much higher compare to the nonsense control.
Fig.1B: It is not clear which shRNA have been used for this analysis.
An important limitation from the live imaging experiments are performed in HEK transfected cells, overexpressing both CXCR4 and LASP1, and stimulated with high dose of CXCL12 (1uM), situation pretty far away from patient´s situation. Authors should comment on this aspect.
Minor points:
Fig.2B: both cell lines display phosphorylation in both Y171 and S146 LASP1. It would be more informative combine the data from all four experiments and show also the ratio between these two phosphorylation sites for each cell lines (interesting it looks inverted).
Fig.2 and 3 and Fig.4 and 5 can be combined. In general the figures are a bit dispersive, trying to compact them in bigger panel may facilitate the flow of the story
Fig.6: please specify that the pictures are made with a confocal microscope
Fig.8A please check by WB the expression of CXCR4 and CXCR7/ACKR3 in both wt and CXCR4-/-
Pag 9, line 248 remove “in”
Author Response
Referee 1
The Manuscript of Butt et al. presents an interesting study describing a novel direct binding of LASP1 to both CXCR4 and Akt1, showing a different phosphorylation state comparing breast cancer and CML cell lines, where CXCR4 plays a quite different role.
All in all, the study is of interesting for the field, reporting a novel LASP1-Akt1 interaction and the comparison of the mechanism in two different tumor entities where CXCR4 has a different role is attractive. However, the quality should be improved adding additional experiments and corrections to make this manuscript suitable for publication in Cells.
Main points:
Why the authors use only these two specific cell lines (MB231 and K562)?
Since the authors claim that this mechanism is generic for BC (breast cancer) and CML, they should use more cells lines to prove that, and not only one for each tumor entity (i.e. at least the first data, Fig.1 and 2, should be performed using several cell lines for both BC and CML to corroborate this statement).
To substantiate the data, we included a more detailed paragraph summarizing comparable effects of LASP1 depletion on AKT phosphorylation in colon cancer, glioblastoma and nasopharyngeal carcinoma recently published by others demonstrating, that the effect of LASP1 on AKT1 is more general.
Fig.1: This WB analysis should be performed also in K562, since the authors are comparing these two different cell lines.
Moreover MDA-MB-231 is a triple negative breast cancer cell line, is this is true for different breast cancer subtype (i.e. luminal)?
We performed the LASP1 knockdown in MCF-7 (a luminal cell line) and in LASP1 depleted K562 cells - with similar results. The data are shown in a new Figure 1.
Fig.1A: Authors should explain why the expression level of LASP1 in all shRNAs – Dox are much higher compared to the nonsense control.
We have no explanation for that effect.
Fig.1B: It is not clear which shRNA have been used for this analysis.
The used MDAMB-231 cell line with inducible shRNA-LASP1 was generated earlier (Ref. 3). This is now stated in the Methods part.
An important limitation from the live imaging experiments are performed in HEK transfected cells, overexpressing both CXCR4 and LASP1, and stimulated with high dose of CXCL12 (1uM), situation pretty far away from patient´s situation. Authors should comment on this aspect.
We agree with the reviewer that 1μM CXCL12 appears to be a high concentration and we are glad to be given the chance to explain our choice of ligand concentration for this experiment in more detail. We have to use such high ligand concentration as a compromise to visualize the protein translocation within the timespan of the imaging experiment. In order to internalize, the receptor has to be occupied by a ligand to trigger the events leading to receptor internalization. Since ligand binding on-rates depend on the ligand concentration, the kinetics of this internalization event depend on the ligand concentration as well and thus work with maximal speed at saturating ligand concentrations. The process also occurs at lower ligand concentration, but it would be more difficult to demonstrate using confocal imaging approaches. Over a much longer experimental timespan required at lower ligand concentrations, photobleaching of the attached fluorophore might become an issue for visualization of the process in question. Currently we still have to use tagged receptors or proteins for visualization that are transfected and therefore will be overexpressed. Unless endogenously expressed receptors fluorescently tagged via CRISPR/Cas become available, we have to accept these experimental limitations and use the appropriate care for the interpretation of the corresponding results.
Minor points:
Fig.2B: both cell lines display phosphorylation in both Y171 and S146 LASP1. It would be more informative combine the data from all four experiments and show also the ratio between these two phosphorylation sites for each cell lines (interesting it looks inverted).
We expanded this phosphorylation experiment and included MCF-7 (a luminal breast cancer cell line) and LAMA-84 a second CML cell line. The ratio is given now and underlines the higher LASP-S146 phosphorylation in breast cancer cells.
Fig.2 and 3 and Fig.4 and 5 can be combined. In general the figures are a bit dispersive, trying to compact them in bigger panel may facilitate the flow of the story
As recommended by the referees, we have combined Figs. 2 and 3 as well as Figs. 4 and 5.
Fig.6: please specify that the pictures are made with a confocal microscope
We specified this in the legend (new Fig. 4).
Fig.8A please check by WB the expression of CXCR4 and CXCR7/ACKR3 in both wt and CXCR4-/-
We, of course, tried that but the native expression levels of both receptors are too low for detection by commercial antibodies. In addition, only one out of four tested CXCR4 antibodies is specific (see also new Ref. 10). That´s why we used PCR to verify expression/deletion.
Pag 9, line 248 remove “in”
We removed the word “in”.
Reviewer 2 Report
CXCR4-LASP1-AKT1 Interaction- Differences between Breast Cancer and CML
The manuscript by Butt and colleagues evaluated the interactions of different phosphorylated states of LASP1 with both CXCR4 and AKT1. They were able to show the preferential direct binding of both AKT1 and CXCR4 to pLASP-S146 and also demonstrated the hinderance of LASP1-S146 phosphorylation by LASP1-Y171 phosphorylation and vice-versa. Importantly, they were able to show that the different phosphorylation states led to different outcomes (phenotypes) when a comparison between breast cancer and chronic myeloid leukemia cells were made. The experiments were carefully and logically executed, and the newly presented information is worthy of publication. However, the following points need to be adequately addressed:
Major Concerns:
It is not quite clear why the authors decided to compare breast cancer with CML. These are two distinct unrelated cancer entities and have different driver molecules and active signaling cascades. For instance, CXCR4 is overexpressed in breast cancer while it is downregulated in CML, in part as a result of BRR-ABL activity. It will be very helpful if the authors could explain the context and reasoning behind the comparison
In order to make valid deductions when making assertions as to the impact of the different phosphorylated states of LASP1, then similar experiments need to be done in both cell lines. For instance, to confirm the effect of LASP1 depletion on AKT1 phosphorylation, the authors conducted experiments in MDA-MB 231 breast cancer cell line and the corresponding experiments in the CML cell line are missing. Likewise, in the colocalization experiments of LASP1 and AKT1 (Figure 6) only the breast cancer cell line was analyzed.
Minor points:
The authors showed a hinderance of pLASP1-S146 by pLASP1-Y171 and vice-versa (Figure 7B). However, in the TKI treatment experiments, the authors showed that increasing pLASP1-S146 phosphorylation with Nilotinib did not have any effect on Y171 phosphorylation and even with Dasatinib a slight reduction was observed. Moreover, a a pradoxical increase in pLASP1-S146 was observed with 12 nM Dasatinib having a weaker effect than the 6nM. Do the authors have a potential explanation?
The decrease in pAKT1-S473 phosphorylation following LASP1 depletion is represented with a line densitogram in Figure 1B. This figure shows 6 lines with –Dox and + Dox changes. What each individual line represents is not delineated
Figure 4 and 5 could be combined
The B-Actin bands in Figure 3A are missing
To counteract the possible effect of cell line specific observations, experiments in an additional cell line for each cancer entity will be useful.
The article title sounds a like a review. It will be nice if the authors modify the title slightly to reflect their important findings
Author Response
Referee 2
Comments and Suggestions for Authors
The manuscript by Butt and colleagues evaluated the interactions of different phosphorylated states of LASP1 with both CXCR4 and AKT1. They were able to show the preferential direct binding of both AKT1 and CXCR4 to pLASP-S146 and also demonstrated the hinderance of LASP1-S146 phosphorylation by LASP1-Y171 phosphorylation and vice-versa. Importantly, they were able to show that the different phosphorylation states led to different outcomes (phenotypes) when a comparison between breast cancer and chronic myeloid leukemia cells were made. The experiments were carefully and logically executed, and the newly presented information is worthy of publication. However, the following points need to be adequately addressed:
Major Concerns:
It is not quite clear why the authors decided to compare breast cancer with CML. These are two distinct unrelated cancer entities and have different driver molecules and active signaling cascades. For instance, CXCR4 is overexpressed in breast cancer while it is downregulated in CML, in part as a result of BRR-ABL activity. It will be very helpful if the authors could explain the context and reasoning behind the comparison
We have included two more sentences and references into the introduction to explain the context behind the comparison of CML and breast cancer:
In this regard, LASP1 has been identified as member of a six genes signature highly predictive for CML (new Ref. 9) and a role of LASP1-CXCR4 in CML progression is discussed (new Ref. 10)
In addition, we investigated the differences in LASP1-CXCR4 signaling in breast cancer and CML as these two tumor entities display different roles for CXCR4 despite comparable overexpression of the receptor binding-partner LASP1.
In order to make valid deductions when making assertions as to the impact of the different phosphorylated states of LASP1, then similar experiments need to be done in both cell lines. For instance, to confirm the effect of LASP1 depletion on AKT1 phosphorylation, the authors conducted experiments in MDA-MB 231 breast cancer cell line and the corresponding experiments in the CML cell line are missing. Likewise, in the colocalization experiments of LASP1 and AKT1 (Figure 6) only the breast cancer cell line was analyzed.
We repeated the AKT -phosphorylation experiments with LASP1-depleated K562 cell lines (and MCF-7 luminal breast cancer as asked for by the second referee) and performed immunofluorescence staining for LASP1 and AKT1 in K562 cells.
Similar to MDAMB-231, knockdown of LASP1 in K562 (and MCF-7) resulted in reduced pAKT-S473 levels (new Fig.1).
As expected, due to high LASP1-Y171 phosphorylation that showed no interaction with AKT1 in vitro, no co-localisation between LASP1 and AKT1 was observed in K562 cells. However, a randomly coexistence in the cytosol because of the large nuclei and the reduced cytoplasm volume is visible (new Fig. 4).
Minor points:
The authors showed a hinderance of pLASP1-S146 by pLASP1-Y171 and vice-versa (Figure 7B). However, in the TKI treatment experiments, the authors showed that increasing pLASP1-S146 phosphorylation with Nilotinib did not have any effect on Y171 phosphorylation and even with Dasatinib a slight reduction was observed. Moreover, a pradoxical increase in pLASP1-S146 was observed with 12 nM Dasatinib having a weaker effect than the 6nM. Do the authors have a potential explanation?
Nilotinib is only inhibiting BCR-ABL while Dasatinib inhibits several tyrosine kinases.
After Nilotinib treatment, other tyrosine kinases like Src and Lyn are still active and continue phosphorylating LASP1. That´s why we still see LASP1-Y171 phosphorylation after Nilotinib incubation. Only after Dasatinib treatment a reduced pLASP-Y171 is observed. This is stated in the manuscript (lines 336 ff).
The decrease in pAKT1-S473 phosphorylation following LASP1 depletion is represented with a line densitogram in Figure 1B. This figure shows 6 lines with –Dox and + Dox changes. What each individual line represents is not delineated
We expanded Fig. 1 with more cell lines and improved the legend as well as the Methods part, now clearly stating the experimental conditions.
Figure 4 and 5 could be combined
Figs. 4 and 5 have been combined
The B-Actin bands in Figure 3A are missing.
The confusing labeling was removed.
To counteract the possible effect of cell line specific observations, experiments in an additional cell line for each cancer entity will be useful.
We repeated the experiments with the luminal MCF-7 cell line and with LAMA-84, a second CML cell line - with similar results (new Figs. 1 and 2).
The article title sounds a like a review. It will be nice if the authors modify the title slightly to reflect their important findings.
The title has changed to:
Phosphorylation-dependent differences in CXCR4-LASP1-AKT1 interaction between breast cancer and CML
Round 2
Reviewer 1 Report
I appreciate the modifications made by Authors, in particular adding more cell lines, and reformatting the figures.
The quality of the study has been improved.